# Recent Advances in the Synthesis of Propargyl Derivatives, and Their Application as Synthetic Intermediates and Building Blocks [note 1]

**DOI:** 10.3390/molecules28083379

**Published:** 2023-04-11

**Authors:** Rodrigo Abonia, Daniel Insuasty, Kenneth K. Laali

**Affiliations:** 1Research Group of Heterocyclic Compounds, Department of Chemistry, Universidad del Valle, Cali A.A. 25360, Colombia; rodrigo.abonia@correounivalle.edu.co; 2Grupo de Investigación en Química y Biología, Departamento de Química y Biología, Universidad del Norte, Barranquilla 081007, Atlántico, Colombia; insuastyd@uninorte.edu.co; 3Department of Chemistry, University of North Florida, 1 UNF Drive, Jacksonville, FL 32224, USA

**Keywords:** propargylating agents, target substrates, catalysts and catalytic systems, propargylated building blocks and intermediates, homopropargylic reagents, application in synthesis

## Abstract

The propargyl group is a highly versatile moiety whose introduction into small-molecule building blocks opens up new synthetic pathways for further elaboration. The last decade has witnessed remarkable progress in both the synthesis of propargylation agents and their application in the synthesis and functionalization of more elaborate/complex building blocks and intermediates. The goal of this review is to highlight these exciting advances and to underscore their impact.

## 1. Introduction

The present review covers relevant literature published from 2010 to present. According to the consulted reports, whereas in the majority of cases the target compounds result from direct introduction of the propargyl moiety, in many examples, the propargylation reaction serves as a strategic step in a reaction sequence that results in the formation of more elaborate/complex structures. In such cases, this review emphasizes the propargylation methodologies rather than the subsequent steps en route to more complex synthetic targets. It is noteworthy that tautomerization between the propargyl (**I**) and allenyl (**II**) moieties (Figure 1) greatly expands the scope of propargylation, since either one may function as a propargylation agent [1,2]. Indeed, in many examples discussed in this review, allenyl derivatives and propargyl derivatives can be employed interchangeably to obtain the same propargylated derivative, or be applied to different substrates, all leading to the propargylated analogs.

As depicted in Table 1, this review is organized based on the type of substrate/functional group reacting with various classes of propargylating reagents (propargyl and/or allenyl derivatives), while also highlighting the catalysts/catalytic systems employed, including complex catalytic systems formed via catalyst/ligand interactions applied to asymmetric propargylation.

## 2. Types of Substrates

### 2.1. (a) Aldehydes and Ketones and (b) Hemiacetals

A propargylation reaction in carbonyl derivatives (aldehydes and ketones) whereby the propargylation reagent acts as a nucleophile toward the C=O functionality is a convenient method for the synthesis of chiral and achiral secondary or tertiary homopropargylic alcohols from aldehydes or ketones, respectively [3]. Significant progress has been made in the development of chiral propargylation reagents and diastereoselective additions of propargylic anion equivalents to chiral aldehydes and ketones [4].

Homopropargylic alcohols are present as fundamental structural entities in many bioactive compounds [5,6], and have also attracted significant interest as useful building blocks for complex molecule synthesis [7,8,9]. In this regard, several synthetic strategies and propargylation reagents have been employed for the synthesis of this interesting family of alcohols, as summarized below.

(a)Aldehyde and ketones

#### 2.1.1. With Boron-Based Propargyl Reagents

Propargyl–/allenyl–boron-based compounds are a family of propargylation reagents with easy availability and relatively low costs, and for this reason, they are widely used in the propargylation processes of diverse organic substrates, as summarized in Table 2 and Figure 2, Figure 3 and Figure 4.

Following the discovery of the highly enantioselective and site-selective copper alkoxide-catalyzed propargylation of aldehydes **1** (R^1^ = H) with a propargyl borolane **2a** (Table 2, entry 1), a catalytic cycle based on a Cu-alkoxide-mediated B/Cu exchange with propargyl borolane **2a** was proposed, with an allenyl Cu intermediate as a key species. Additional experiments demonstrated the proposed catalytic cycle [10]. Table 2 also summarizes several other synthetic approaches to the propargylation reaction of diverse aldehydes and ketones **1** through propargyl/allenyl borolane reagents **2**, producing a variety of chiral and achiral secondary and tertiary homopropargylic alcohols **3**.

A simple protocol for the synthesis of homopropargyl alcohols **5**, starting with isatin derivatives **4** under mild reaction conditions, was reported (Figure 2) [22]. Reactions were performed in the presence of copper triflate as a Lewis acid catalyst, with allenylboronic acid pinacol ester **2c** as a nucleophile, in aqueous media, producing excellent product **5** yields. The enantioselective synthesis of chiral propargyl alcohols **6** was also explored. The best regioselectivity was achieved when (*S*)-SEGPHOS was used as a chiral ligand, resulting in enantiomeric ratios up to 12:88. Gram-scale synthesis, performed to check the efficiency of the protocol, showed retention in selectivity [22].

The synthesis of tri- and tetrasubstituted allenylboronic acids was established via a versatile copper-catalyzed methodology (Figure 3) [23]. Subsequently, the obtained allenylboronic acids **7** were subjected to propargylboration reactions with ketones **1** without any additives, producing homopropargyl alcohols **8** (Figure 3). Additionally, catalytic asymmetric propargylboration of the ketones **1** with high stereoselectivity was achieved when (*S*)-Br_2_-BINOL was used as chiral ligand, allowing for the synthesis of highly enantioenriched tertiary homopropargyl alcohols **9** (Figure 3). The reaction was suitable for the kinetic resolution of racemic allenylboronic acids, producing alkynes with adjacent quaternary stereocenters [23].

The propargylation of aldehydes/ketones **1** using potassium allenyltrifluoroborate **10** promoted by tonsil, an inexpensive and readily available clay, in a chemo- and regioselective manner was described, leading to homopropargyl alcohols **11** in good to moderate yields (Figure 4, entry 1) [24]. The described method is simple and avoids the use of air- and moisture-sensitive organometallics. In the same way, alcohols **11** were synthesized under MW irradiation (Figure 4, entry 2) [17] or by using Amberlyst A-31 (Figure 4, entry 3) [25].

#### 2.1.2. With Propargyl Silanes

In the context of silane-mediated transformations promoted by chiral Lewis base catalysis, it has been shown that the coupling of a Lewis base with a silane reagent can promote several synthetically useful reactions, opening up the possibility for further studies [26]. In a recently developed catalytic asymmetric addition process (Figure 5), optically active homopropargylic alcohols **13** were synthesized by reacting propargylic silanes **12** with aldehydes **1** (R = H), using a chiral organosilver species as a pre-catalyst. The catalyst was formed in situ via an (*R*)-DM-BINAP⋅AgBF_4_ complex. The other additives were TEA (base pre-catalyst), along with KF and MeOH [27].

Allenyltrichlorosilane is an attractive candidate as a nucleophilic partner in C=O and C=N propargylation reactions because of its mildness, regiospecificity, and low toxicity [28]. It was reported that a new bidentate helical chiral 2,2′-bipyridine *N*-monoxide Lewis base can efficiently catalyze the addition of allenyltrichlorosilane **14** to aromatic aldehydes **1** (R = H), producing homopropargylic alcohols **15** with high levels of enantioselectivity and high yields (Figure 6, entry 1) [29]. Additionally, extensive computational studies have made it possible to predict stereoselectivities for the synthesis of alcohols **15** using axially chiral bipyridine *N*,*N*’-dioxides as catalysts (Figure 6, entries 2 and 3). It was found that the stereoselectivity of these bidentate catalysts is controlled by well-defined rigid transition-state structures. It was suggested that *N*,*N’*-dioxides are superior platforms for rational catalyst development for asymmetric propargylation [30,31].

Xanthones, thioxanthones, and xanthenes are naturally occurring molecules and have interesting properties due to their special structures [32,33]. With this in mind, gold-catalyzed bispropargylation of xanthones and thioxanthones **16** (X = O, S, respectively) was devised (Figure 7) [34]. In this approach, the use of propargylsilanes **17** permitted deoxygenative bispropargylation through the double catalytic addition of the corresponding allenylgold intermediate to the synergistically activated carbonyl moiety. This methodology worked in a diastereoselective manner, with either xanthone or thioxanthone derivatives **16**, producing the corresponding 9,9-bispropargylxanthenes and thioxanthenes **18** (X = O, S, respectively) in high yields.

#### 2.1.3. With Propargyl Halides

The addition of organochromium reagents to carbonyl compounds is considered an important tool in contemporary organic synthesis because of a number of unique features, such as mild reaction conditions, high chemoselectivity, and compatibility with a wide range of functional groups [35]. Chiral homopropargyl alcohols **3** were envisioned among the products potentially accessible using this methodology. Most of the asymmetric methods that provide access to these compounds involve the use of chiral allenyl reagents, for which catalytic enantioselective NH propargylation was considered a suitable alternative, owing to the ready availability of propargyl halides **19** as sources of propargyl moieties.

Following the development of a tethered *bis-*(8-quinolinato) (TBOx) chromium complex [36], it was successfully used as a highly stereoselective catalyst for several asymmetric reactions [37,38,39,40]. Its application as a catalyst was extended to the asymmetric NH propargylation of aldehydes. Thus, a highly enantioselective catalytic system for the NH propargylation of aldehydes **1** (R = H) via a Barbier-type reaction [41] employing low Mn catalyst loading was developed (Table 3, entry 1). High enantioselectivities, not previously achievable for aromatic, heteroaromatic, and α,β-unsaturated aldehydes using NH chemistry, were reported for a range of substrates **1** [42].

Several other approaches to the synthesis of diversely substituted chiral and achiral homopropargyl alcohols **3**, starting with carbonyl compounds **1** and employing halogen-based propargylation reagents **19**, in the presence of a variety of catalytic systems, are outlined in Table 3 and Figure 8.

A protocol for the total synthesis of (–)-epiquinamide involving the L-proline-catalyzed one-pot sequential α-amination/propargylation of aldehyde **1** (R = H) was established (Figure 8). The synthesis was accomplished in nine steps, with the formation of homopropargyl alcohol **20** as a strategic step (entry 1) [48]. In the same way, six-step asymmetric total synthesis of the natural pyrrole lactone longanlactone was designed. The reaction involved the formation of propargyl alcohol **22** through the Zn-catalyzed Barbier propargylation of the aldehyde **21** as one of the key steps in this process (Figure 8, entry 2) [49].

A chemo-enzymatic process was established as a useful method for the derivatization of galactose unit of spruce galactoglucomannan (GGM) and other galactose-containing polysaccharides. In this approach, a series of GGMs were selectively formylated at the C-6 position via enzymatic oxidation by galactose oxidase. The formed aldehydes **23** were further derivatized via an indium-mediated Barbier–Grignard-type reaction using propargyl bromide **19a**, resulting in the formation of homoallylic alcohols **24** (Figure 8). All the reaction steps were performed in water in a one-pot reaction. The formation of the propargylated products was identified via MALDI-TOF–MS. The polysaccharide products were isolated and further characterized via GC–MS or NMR spectroscopy. The derivatized polysaccharides **24** were considered potential platforms for further functionalization (entry 3) [50].

A stereospecific Barbier-type reaction of α-hydroxyketones **25** with propargyl bromide **19a** in the presence of indium metal provided (1*RS*,2*SR*)-1,2-diarylpent-4-yne-1,2-diols **26** in good yields as single diastereomers (Figure 8). The observed high diastereoselectivity (>99%) in 1,2-diols **26** was consistent with the Cram’s chelation model [51]. The 1,2-diols **26** were successfully used as precursors for furan synthesis through iodine-mediated 5-*exo*-*trig* cyclization, dehydration, and reductive deiodination (entry 4) [52].

Another study described diastereoselective Zn-mediated propargylation for non-enolizable norbornyl α-diketones **27**. In this approach, the treatment of **27** with zinc and propargyl bromide **19a** in anhydrous THF, using the Barbier procedure under ultrasound, produced the corresponding norbornyl homopropargyl alcohols **28** in good yields (Figure 8). An analysis of the crude reaction mixtures revealed that **28** was obtained in a diastereomerically pure form, along with small amounts of allene derivatives as byproducts. Moreover, the stereochemistry of **28** was confirmed via X-ray crystal structure analysis. Subsequently, homopropargyl alcohols **28** were used as precursors for an AgI-catalyzed cycloisomerization toward diversely substituted spirocyclic dihydrofuran derivatives and produced acceptable to good yields (entry 5) [53].

Based on the dual photoredox catalytic strategy [54,55], practical and effective photoredox propargylation of aldehydes **1** (R = H) promoted by [Cp_2_TiCl_2_] was developed (Figure 9). The reaction did not require stoichiometric metals or scavengers, and employed a catalytic amount of [Cp_2_TiCl_2_], along with the organic dye 3DPAFIPN (as a reductant for titanium). The reaction displayed a broad scope, producing the desired homopropargylic alcohols **29** in good yields with both aromatic and aliphatic aldehydes [56].

The synthesis of homopropargyl alcohol **31** with a two-carbon extension was achieved through the propargylation of aldehydes **1**, mediated by zinc(0). This reagent was generated in situ from the redox coupling of Al and ZnCl_2_ in 2N HCl and THF, producing products **31** in acceptable to good yields (Figure 10) [57].

Aldehydes **1** were transformed into their corresponding homopropargyl alcohols **32** via a reaction with propargyl bromide **19a**, with CuCl and Mn powder employed in the presence of TFA in ACN solvent (Figure 11). This method proved compatible with a variety of substrates, leading to diversely substituted products **32** in high yields. A large-scale reaction was also performed, demonstrating the potential synthetic applications of this transformation [58].

#### 2.1.4. With Organometallic Propargyl Reagents

The Barbier type nucleophilic addition of functionalized halides to carbonyls mediated by metals or metal compounds constitutes an important strategy for carbon–carbon bond formation in organic synthesis [59,60,61]. In this context, an operationally simple procedure for the propargylation of aldehydes **1** in moist solvent (distilled THF) was developed through the direct addition of propargyl bromide **19a** to the aldehyde substrates **1**, mediated by low-valent iron or tin (Figure 12). The metals were prepared in situ using a bimetal redox strategy. Using different aldehydes **1** as substrates, both metals proved applicable, producing homopropargyl alcohols **34** in good yields and with high chemoselectivity in most cases. Due to its efficacy, operational simplicity, performance in moist solvent, and its use of inexpensive metal/metal salts, the procedure was claimed to be practically viable and potentially scalable [62].

Allenyl boronic acids are widely used as propargylation reagents. These compounds are usually prepared via the Hg-catalyzed magnesiation of propargyl bromide [63]. However, the use of mercury, the corrosiveness of propargyl bromide, and the pyrophoric nature of allenyl boronic acid raise environmental and safety concerns, particularly when using these reagents for large-scale applications. To circumvent these limitations, the development of a mercury-free flow chemistry process for the asymmetric propargylation of aldehydes using allene gas **35** as a reagent was reported (Figure 13). The connected continuous processes of allene dissolution, lithiation, Li-Zn transmetalation, and the asymmetric propargylation of the chiral aldehyde **38** provided a homopropargyl β-amino alcohol **39** with high regio- and diastereoselectivity in high yield. This flow process represents a practical use for an unstable allenyllithium intermediate **36**, using the commercially available and recyclable (1*S*,2*R*)-*N*-pyrrolidinyl norephedrine (L*) as a ligand to promote the diastereoselective propargylation of **38** [64].

The esters of 4-hydroxybut-2-ynoic acid (alkyl 4-hydroxybut-2-ynoates) **42** are promising building blocks for organic synthesis. The presence of three important functional groups, namely the acetylene bond conjugated with the ester moiety, and the hydroxyl group of the propargyl unit in the structure of these compounds, make them highly versatile and applicable to many useful synthetic transformations [65,66,67,68,69,70]. With this in mind and based on previous works on the superelectrophilic activation of acetylene compounds [71], a series of 4-aryl(or 4,4-diaryl)-4-hydroxybut-2-ynoates **42** were obtained for further studies on their transformations under the action of various acids. The treatment of propynoates **40** with a solution of BuLi in hexanes produced lithiated intermediates in situ **41**. Then, carbonyl compounds **1** were added at low temperature to form the target alkyls 4-hydroxybut-2-ynoates **42** in acceptable to excellent yields (Figure 14) [72].

Epoxides serve as both building blocks and synthetic intermediates in various organic transformations [73,74]. The conjugation of a propargyl group to an epoxide creates a highly functional small-molecule building block. A series of substituted propargyl epoxides **45** were prepared via the propargylation of α-bromoketones **43** with an organozinc reagent **44** (Figure 15). This method complements existing synthetic methods due to the advantageous properties of the organozinc reagents, such as their availability, selectivity, operational simplicity, and low toxicity [75].

#### 2.1.5. With Propargylic Ethers, Acids, and Esters

The intramolecular propargylation of aldehydes and ketones enables their entry into cyclic compounds containing a homopropargyl alcohol unit, a structural motif that is present in a variety of biologically active compounds and is highly useful for synthetic transformations [76,77]. Due to their ready availability, propargylic esters **46** [78] are logical starting points in these transformations. It has been shown that carbonyl-tethered propargylic benzoates **46** undergo intramolecular carbonyl propargylation upon treatment with Et_2_Zn in the presence of a catalytic amount of Pd^0^ to form 2-alkynylcyclopentanol products **47** (Figure 16). Diastereoselectivity for the formation of simple homopropargylcycloalkanols **47**, generated through the use of Pd^0^/Et_2_Zn, was examined as a function of the palladium phosphine ligand in the absence of further structural constraints imposed by additional substituents or rings. In this approach, a ligand/solvent effect on the *cis/trans* selectivity (referring to the relative positions of the alkynyl and OH groups) of ring-closure was found. In a non-coordinating solvent (benzene), increasing the electron-donating ability of the phosphine ligand (while decreasing its dissociation ability) led to an increased tendency towards the *trans* product, while the combination of a coordinating solvent (THF) and PPh_3_ resulted in the exclusive formation of *cis* products. The experimental and computational results were compatible with the divergent behavior of an allenyl-ethylpalladium intermediate that partitions between competitive carbonyl-addition and transmetalation pathways, each leading to a different diastereoisomers. The results also suggested that the dissociating ability of the phosphine acted as a regulating factor for this behavior [79].

Isolated in 2008 from the marine sponge *Siliquariaspongia mirabilis*, mirabalin [80] was found to inhibit the growth of the tumor cell line HCT-116, with an IC_50_ value of 0.27 μM. This compound belongs to the chondropsin family of macrolide lactams, which comprises chondropsins A−D, 73-deoxychondropsin A, and poecillastrins A−C [81]. Alcohol **50** is a key intermediate in the convergent and flexible stereoselective synthesis of one isomer of the C44−C65 fragment of mirabalin [82]. To synthesize alcohol **50**, aldehyde **48** was subjected to stereoselective Marshall allenylation [83] through the addition of a chiral allenylzinc reagent, prepared in situ via palladozincation of the (*S*)-propargylic mesylate **49**. This method delivered propargyl alcohol **50** with good diastereoselectivity in favor of the *anti*,*syn*,*anti*-isomer (Figure 17). The two diastereomers were separated via flash chromatography on silica gel.

The transition metal-catalyzed carbonyl propargylation protocol is an elegant approach to the diastereo- and enantioselective construction of homopropargylic alcohols. Addition reactions of propargyl metal or metalloid to aldehydes have been widely used as general synthetic methods. Nevertheless, some limitations exist in this strategy because of its ambident nucleophile characteristics as propargyl/allenyl organometallic reagents, which open up new reaction channels and widen their synthetic scope [84,85]. To circumvent these limitations, researchers have focused on transition metal-free carbonyl propargylation for the synthesis of 1,2,4-substituted homopropargylic alcohols.

In this regard, a transition metal-free three-component process was developed by combining aldehydes **1**, 3-(tributylstannyl)propargyl acetates **51** formed in situ from readily available propargyl acetates, and trialkylboranes **52**, providing access to a range of 1,2,4-trisubstituted homopropargylic alcohols **53** (Figure 18). It was found that the addition of diisopropylamine played a crucial role in the selective formation of homopropargylic alcohols **53**. Importantly, this methodology could be extended to a single-flask reaction sequence starting with propargyl acetates [86].

Although propargylic carbonates are readily available compounds that could potentially be used instead of the corresponding propargylic halides in the carbonyl propargylation process, they are inert under classical Barbier conditions. Whereas notable examples of the use of propargyl carbonates have been described, their applications were typically limited to aldehydes as electrophiles [78,87]. To circumvent this limitation, an efficient protocol for the synthesis of homopropargylic alcohols **55** in moderate to good yields was reported that utilized propargylic carbonates **54** as pronucleophiles (Figure 19). This reaction is based on a combination of transition metal (palladium) and radical (titanium) chemistry, in which allenyl titanocenes and transient propargylic radicals are formed in situ as key species for the success of this multimetallic protocol. The reaction took place with excellent regioselectivity, tolerating a variety of terminal and internal alkyne functionalities of the starting propargylic carbonates **54** with different substitution patterns, as well as diverse carbonyl compounds **1** (aldehydes and ketones), thus providing a useful method for application in synthetic organic chemistry (entry 1) [88].

In a similar way, low-valent indium(I)-mediated nickel-catalyzed propargylation of aldehydes **1** with propargylic carbonates **54** was established. In this approach, the nickel/indium(I)-mediated reaction of the starting materials **54**, which possessed different substitution patterns, produced *syn*-homopropargylic alcohols **56** in acceptable to high yields upon coupling with a variety of carbonyl compounds **1** (Figure 19). Both the nickel catalyst and the phosphane ligands were found to play a crucial role in this transformation. Diastereoselectivity was also strongly dependent on the ligand employed. Moreover, a mechanistic sequence involving an umpolung of propargylnickel intermediates under the influence of low-valent indium was proposed, to account for the dependence of the stereochemical characteristics of the phosphane ligands (entry 2) [89].

#### 2.1.6. With Methylene-Active Propargyl Compounds

Despite extensive studies on gold catalysis, σ-allenylgold species have not been invoked as catalytic intermediates and their reactivities remain to be studied. In a recent study, the formation of an in situ-generated σ-allenylgold was proposed via soft propargylic deprotonation of the methylene-active derivatives **57**, mediated by the isomerization of an alkyne to an allene. The σ-allenylgold species formed from **57** underwent nucleophilic addition to the activated aldehydes **1** in bifunctional biphenyl-2-ylphosphine (**L1**) ligand-enabled gold catalysis. This development revealed a broad range of opportunities to achieve the propargylic C−H functionalization of **57** under catalytic and mild conditions, producing homopropargyl alcohol intermediates **58** (Figure 20). Subsequently, the resulting homopropargyl alcohols **58** underwent ligand-enabled cycloisomerization, involving an unexpected silyl migration process, to deliver dihydrofurans **59** as isolated products [90].

#### 2.1.7. With 1,3-Enynes

While most methods for enantioselective carbonyl propargylation promote the formation of the parent α-unsubstituted homopropargylic alcohols, less attention has been devoted to the development of diastereo- and enantioselective propargylation protocols that generate useful (α-methyl)homopropargyl alcohols [91]. Under the conditions of ruthenium-catalyzed transfer hydrogenation, employing isopropanol as a source of hydrogen, unprotected isopropoxy-substituted enyne **60** and aldehydes **1** engaged in reductive coupling to provide propargylation product (α-methyl)homopropargyl alcohols **61** with good to complete levels of anti-diastereoselectivity (Figure 21). Remarkably, it was found that the unprotected tertiary hydroxy moiety of isopropoxy enyne **60** is required in order to enforce diastereoselectivity. Moreover, deuterium-labeling studies corroborated reversible enyne hydrometalation in advance of carbonyl addition. Additionally, it was demonstrated that the isopropoxy group of products **61** could be readily cleaved upon exposure to aqueous sodium hydroxide to reveal the terminal alkyne functionality [92].

#### 2.1.8. With Aryl-Acetylenes

The Favorskii reaction, which involves the nucleophilic addition of alkynes to aldehydes in the presence of a strong base, has been recognized as an efficient synthetic strategy to produce propargyl alcohols and α,β-unsaturated ketones [93]. Direct propargylation/alkenylation via the allenol-enone isomerization sequence through the activation of the C-H bond in terminal alkynes, without a transition metal and employing a weak base, represents a challenging research area. In response to this, a fast and efficient transition metal-free, modified Favorskii-type direct alkynylation protocol for the synthesis of propargyl alcohols **63/65** was developed using a combination of Cs_2_CO_3_ and TEA as weak bases (Figure 22). Aliphatic aldehydes **1** (R^1^ = H) produced propargyl alcohols **63**, while cyclic ketones **64** furnished propargyl alcohols **65.** The operationally simple protocol, wide substrate scope, and gram-scale synthesis represent key aspects of this methodology. A plausible mechanism for this transformation involving the weak base-assisted propargylation of carbonyl compounds **1** was suggested [94].

(b)Hemiacetals

The development of copper(I)-catalyzed stereodivergent anomeric propargylation of unprotected aldose **66** was established as a facile synthetic pathway to a broad variety of sialic acid derivatives **69**, via a key propargylation intermediate **68** (Figure 23). The reaction proceeded with the in situ formation of a soft allenylcopper(I) species, catalytically generated from the stable allenylboronic acid pinacolate **2c**. It was also observed that the addition of B(OMe)_3_ facilitated the ring-opening of the non-electrophilic cyclic hemiacetal form of aldose **66** to reach its corresponding open-chain reactive aldehyde form **67**, subsequently leading to the formation of the key intermediate **68**. This synthetic method, which required no protecting groups, could be performed at the gram-scale, offering general and practical access to various sialic acid derivatives from unprotected-type aldoses **66** [95].

In a similar way, copper(I)-catalyzed stereodivergent nucleophilic propargylation at the anomeric carbon of unprotected *N*-acetyl mannosamine **70** was devised using 3-substituted allenylboronates **2c** as nucleophiles (Figure 24). The homopropargylic alcohol products **71** and **72** containing two contiguous stereocenters, and two stereoisomers out of the four possible isomers, were selectively obtained in a catalyst-controlled manner by applying either basic conditions (a MesCu/(*R*,*R*,*R*)-Ph-SKP catalyst with a B(O*i*Pr)_3_ additive) or acidic conditions (a CuBF_4_/(*S*,*S*,*S*)-Ph-SKP catalyst with an MeB(O*i*Pr)_2_ additive). In the following two steps, the propargylation products **71** and **72** were transformed into C3-substituted sialic acids without the use of protecting groups [96].

### 2.2. (a) Imines, (b) Iminium, and (c) Azo Compounds

(a)Imines

The addition of organometallic reagents to imines is one of the most useful and versatile methodologies for creating both a new carbon–carbon bond and new amine functionality [97]. When a propargyl organometallic reagent is used [98], via diverse synthetic strategies, the process offers the possibility for further transformation of the unsaturation to form more carbon–carbon or carbon–heteroatom bonds [99], thus giving practical use to this synthetic approach.

#### 2.2.1. With Propargyl Halide/Metal Reagents

The enantio- and/or diastereoselective version of the propargylation of imines is of additional interest because at least one new stereogenic center is created [100]. Moreover, α- or γ-substitution in the imine reagent could also induce chemoselectivity in this process because the propargyl moiety could be selectively added to the structure of the product [101]. Using this approach, the diastereoselective Barbier-type addition of allyl halides to chiral sulfinylimines **73**, promoted by indium metal [102], resulted in the formation of chiral *N*-protected homoallylic amines in good yields and % dr. More specifically, the reaction of different chiral imines **73**, derived from aldehydes or ketones, with the silylated propargyl bromide **19a** under sonication, in the presence of indium metal, led mainly or exclusively to the formation of protected homopropargylamines **74** in a diastereoselective manner (Figure 25, entry 1). Of special interest in this process are the ketimine derivatives **73** (derived from ketones) because the new stereocenter has a quaternary configuration. Further, selective deprotection of the two protecting groups (TMS and sulfinyl moieties) was accomplished using conventional methods [103].

In another approach, a highly efficient method for the asymmetric synthesis of a wide range of quaternary carbon-containing homopropargylic amines **74** via the Zn-mediated asymmetric propargylation of *N*-*tert*-butanesulfinyl ketimines **73** was reported (Figure 25, entry 2). In this approach, the ketimines **73** were readily prepared according to known procedures [104], producing products **74** in good yields and with high diastereoselectivities [105].

A series of enantioenriched homopropargylic amines **74** were obtained in good yields and with excellent diastereomeric ratios via the indium-mediated *N*-propargylation of chiral *N*-*tert*-butanesulfinyl ketimines **73** using trimethylsilylpropargyl bromide **19a**, in the presence of indium metal, under sonication (Figure 25, entry 3). Further, the chiral amines **74** were used as starting materials to obtain access to 3-substituted 1,2,3,4-tetrahydroisoquinoline derivatives in their enantioenriched form [106].

A Zn-mediated propargylation/lactamization cascade reaction with chiral 2-formylbenzoate-derived *N*-*tert*-butanesulfinyl imines **73** (R = aryl, R^1^ = H) was realized, as described in Figure 26. In this strategy, sulfinyl amines **75** were obtained as intermediates, providing a practical and efficient method for the synthesis of chiral isoindolinones **76**. Moreover, high diastereoselectivities and good reaction yields were observed for the majority of the examined cases [107].

An efficient approach to the synthesis of α,α-bispropargyl-substituted amines **78** in acceptable yields was achieved via Zn-promoted aza-Barbier-type reactions of *N*-sulfonyl imidates **77** with various propargyl reagents **19a** (Figure 27, entry 1). The synthetic utility of this approach was demonstrated via the rapid construction of pyrrolidine derivatives [108]. In a similar way, a one-pot method for the synthesis of homopropargylic *N*-sulfonylamines **79** from aldehydes catalyzed by zinc powder was described. The imine derivatives **77** were obtained in situ as intermediates from a reaction between the corresponding aldehydes **1** and TsNH_2_ in the presence of BnBr and Zn. This procedure offers simplicity, good yields, and was shown to be applicable to a variety of aldehydes (Figure 27, entry 2) [109].

The synthesis of 3-propargylated 3-aminooxindoles **81** was carried out via the zinc-mediated propargylation of isatin-derived imines **80** (Figure 28). This approach avoided the use of catalysts, severe reaction conditions, multistep procedures, and reaction additives. To demonstrate its synthetic utility, different isatin-derived imines **80** and propargyl bromide **19a** were used to obtain products **81** in good yields [110].

#### 2.2.2. With Propargyl/Allenyl Boron Reagents

Expanding the available methods for the synthesis of homopropargylic amines, zinc-catalyzed diastereoselective propargylation of *tert*-butanesulfinyl imines **73** using propargyl borolanes **2a** was reported (Figure 29, entry 1). This method produced both aliphatic and aryl homopropargylic amines **74** in acceptable to good yields and with good stereoselectivity. The utility of the homopropargylic amines **74** was demonstrated in the synthesis of a *cis*-substituted pyrido-indole through diastereoselective Pictet-Spengler cyclization [111].

Allenylborolane **2c** (instead of propargyl borolane **2a**) was employed in the enantioselective Ag-catalyzed propargylation of *N*-sulfonylketimines **82** (Figure 29, entry 2). The reaction was compatible with a wide variety of diaryl- and alkylketimines **82**, producing their respective homopropargylic sulfonamides **83** in high yields and in excellent enantiomeric ratios. It was also found that both propargyl and allenylborolane reagents (**2a** and **2c**) could be used to obtain homopropargylic products **83**, and a mechanism involving transmetalation of the borolane reagent **2c** with a silver catalyst was proposed. Further, the homopropargylic products **83** were used as starting materials to elaborate diverse products of higher complexity with high stereochemical fidelity, including enyne ring-closing metathesis, Sonogashira cross-coupling, and reduction reactions [112].

The catalytic asymmetric propargylation of 3,4-dihydro-β-carboline **84** with allenylborolane **2c** (instead of propargyl borolane **2a**) was investigated (Figure 29, entry 3). Optimization of the reaction conditions in the presence of CuCl and (*R*)-DTBM-SEGPHOS ligands gave chiral scaffolds **85** with reproducible results, good yields, and high *ee* values. Further transformations of **85** via designed Au(I)/Ag(I)-mediated 6-*endo*-*dig* cyclization directly delivered the indolenine-fused methanoquinolizidine core of the akuammiline alkaloid strictamine in its native oxidation state [113].

The copper-catalyzed asymmetric propargylation of cyclic aldimines **86** was also reported. Asymmetric propargylation of a diverse series of *N*-alkyl and *N*-aryl aldimines **86** with propargyl borolanes **2a** was achieved, producing the corresponding chiral propargylamine scaffolds **87** with good to high asymmetric induction (Figure 29, entry 4). The utility of products **87** was further demonstrated via titanium-catalyzed hydroamination and reduction to generate the chiral indolizidines (−)-crispine A and (−)-harmicine alkaloids. Moreover, the influence of the trimers of imines **86** on inhibiting the reaction was identified, and equilibrium constants between the monomers **86** and their trimers were determined for general classes of imines [114].

#### 2.2.3. With Propargyl/Allenyl-MX reagents

The diastereoselective synthesis of enantiopure homopropargylic amines **74** via the propargylation of various *N*-*tert*-butylsulfinylimines **73** with 1-trimethylsilyl allenylzinc bromides **88** was achieved (Figure 30, entry 1). In this approach, the full conversion of imines **73** was observed when two equivalents of Zn derivatives **88** were used, giving homopropargylic amines **74** as single isomers in very good isolated yields [115].

The fluorinated analogs of *tert*-butanesulfinyl imines **73** were considered convenient precursors for a synthetic route to obtaining enantioenriched fluorinated monoterpenic alkaloid analogues via a Pauson–Khand cyclization reaction [116]. In this approach, diastereoselective propargylation of **73** was implemented as the key step to introducing the chiral information necessary for the rest of the synthetic sequence to be performed. In the first assay, the addition of propargyl magnesium bromide **89** to sulfinyl imine **73** (R = CF_3_) in DCM resulted in the formation of homopropargylamine **74** (R = CF_3_) with low diastereoselectivity. When DCM was replaced with THF, not only was the diastereoselectivity vastly improved, but the major diastereoisomer was actually the opposite of the one observed in DCM. Following the latter reaction conditions, sulfinyl amines **74** were obtained in good yields with high diastereoselectivity (Figure 30, entry 2).

The dramatic effect of the solvent in this type of transformation was attributed to differing transition states depending on the nature of the solvent, but it was also suspected that the strong electron-withdrawing characteristics of the fluorinated groups of substrates **73** played a role in increasing the reactivity of the imines **73** and decreasing the difference in energy between the two transition states in non-coordinating solvents such as DCM [116].

#### 2.2.4. With Imino-Masked Propargyl Reagents

Whereas the development of methods for the α-alkylation of carbonyl compounds has advanced tremendously in recent years, catalytic enantioselective α-propargylation is relatively less developed [117,118]. In response to this, a two-step reaction sequence for the asymmetric formal α-propargylation of ketones was introduced (Figure 31). This approach took advantage of the amino-catalyzed conjugate addition of ketones to alkylidene isoxazol-5-ones, producing intermediates **90/91**, which, through a controlled nitrosative degradation event, produced α-propargyl ketones **92/93** in moderate to good yields, with perfect diastereocontrol, good to excellent enantioselectivity, and broad structural scope [119].

(b)Iminium Compounds

#### 2.2.5. With Propiolic Acids

Thermal-induced transition metal-catalyzed decarboxylative coupling reactions are recognized as a powerful tool in organic synthesis and medicinal chemistry as they require simple operation and produce CO_2_ as a byproduct [120,121,122]. Based on previous works in which dipropargylic amines were obtained as side products mediated by isobutylboronic acid reagents [123], the expansion of this chemistry led to the development of a more flexible approach for the synthesis of dipropargylic amines from primary amines, formaldehyde, and propiolic acids under metal-free conditions. After assaying different reaction conditions, a method in which a mixture of amine **94** (R^1^ = H), formaldehyde, and propiolic acid **95** in DCE was heated in a sealed tube produced optimal yields of the target dipropargylic amines **96** (Figure 32). The method exhibited a broad range of functional group compatibility for primary amines **94** and propiolic acids **95**, and produced the corresponding products **96** in low to excellent yields [124].

#### 2.2.6. With Acetylene Derivatives

A series of *N*-heterocyclic silylene-stabilized monocoordinated Ag(I) cationic complexes weakly bound to free arene rings (C_6_H_6_, C_6_Me_6_, and C_7_H_8_) were synthesized, and the efficacy of these electrophilic Ag(I) complexes as catalysts was investigated toward A^3^-coupling reactions, producing a series of propargylamines **97** in good to excellent yields in a tricomponent reaction of amines **94**, acetylenes **62**, and polyformaldehyde (Figure 33). The process was accompanied by the in situ formation of an iminium species from **94** and polyformaldehyde. The best results were obtained when catalyst **A** was used, with low catalyst loading under solvent-free conditions [125].

A library of *N*-propargyl oxazolidines and *N*,*N*-dipropargyl vicinal amino alcohols was prepared through a multicomponent reaction of formaldehyde, β-aminoalcohols **98**, and acetylenes **62** using a copper-catalyzed A^3^-type-coupling process (Figure 34). Whereas the presence of bromide and chloride ions accelerated the process toward open-ring alkynylation, producing dipropargylated products **99**, the presence of the catalytic system Cu/I favored the formation of propargyl oxazolidines **100** [126].

(c)Azo compounds

#### 2.2.7. With Propargyl Halides

The addition of propargylic or allenylic metal reagents to azo compounds is a convenient method for the preparation of propargylic hydrazines [127,128]. Expanding on earlier studies, the Barbier-type propargylation of azo compounds **101** with propargylic halides **19** that utilizes reactive barium as a low-valent metal in THF as solvent was reported (Figure 35), providing diverse propargylic hydrazines **102** regioselectively in moderate to high yields. The corresponding α-adducts **102** were exclusively formed not only from azobenzenes (diaryldiazenes) but also from dialkyl azodicarboxylates. The method was also applicable to γ-alkylated and γ-phenylated propargylic bromides **19**. Notably, the ester moieties of dialkyl azodicarboxylates remained unaffected by the barium reagent, thus providing the corresponding propargylated compounds **102** as unique products [129].

### 2.3. Aryl and Heterocyclic Derivatives

(a)Aryl derivatives

#### 2.3.1. With Propargyl-TMS

Haloarenes are of great synthetic interest, since they are used as structural scaffolds of different compounds employed in catalytic chemistry, medical chemistry, and agrochemistry. Due to this, new strategies have emerged to obtain various halogenated aromatics, for example, the insertion of a substituent in the *ortho*-position with respect to a pre-existing halogen group. In this context, the synthesis of *ortho*-propargyl iodobenzenes **104** represents a desirable goal. A viable procedure to synthesize these derivatives involves reacting (diacetoxyiodo)arenes **103**, previously activated with BF_3_, with a propargyl metalate **12** using an ACN/DCM mixture as solvent, to furnish *ortho*-propargyl iodobenzenes **104** in moderate to high yields (Figure 36), as described in [130]. A striking feature of this protocol is that it generates a singly propargylated product **104** for each substrate **103** bearing a single type of *ortho*-CH site. The regioselectivity is affected by the electronic environment of the iodoarene nucleus **103**, and the method is applicable to electron-deficient iodoarenes **103**.

Synthetic access to *ortho*-propargylated arylsulfides, as in compounds **106**, is also of great interest, since a variety of synthetic derivatives with a wide catalog of applications can be produced from these types of structures. Compounds **106** have been synthesized in good to excellent yields via a cross-coupling reaction between aryl-sulfoxide **105** and propargylsilanes **17**, using Tf_2_O as an electrophilic activator and 2,6-lutidine as base in ACN (Figure 37). The addition of 2,6-lutidine improved their reaction yields and prevented the formation of undesirable products via acid-mediated cyclization. A plausible mechanism for this metal-free cross-coupling process involves an interrupted Pummerer/allenyl thio-Claisen rearrangement, where the formation of classic Pummerer products did not occur, even in the presence of electron-scavenging alkyl chains on sulfur. Hence, this methodology allows for the formation of sp^2^-sp^3^ C-C bonds in products **106** in an efficient and regioselective manner [131].

#### 2.3.2. With Propargyl Alcohols

The nucleophilic substitution of the -OH group in propargyl alcohols is an efficient methodology for the preparation of synthetic precursors, which, due to its versatility, could be further implemented in synthetic schemes via alkyne functionality and the possible addition of acetylides to different carbonyls. However, this type of substitution is challenging in aryl-propargyl alcohols due to the low reactivity of the hydroxyl as a leaving group and the formation of unwanted side products, as well as polymers originating from unstable/highly reactive carbocationic intermediates. The viable alternative methods for the preparation of propargyl derivatives, such as **108**, via the nucleophilic substitution of aryl-propargyl alcohols **63** are highlighted in Figure 38.

There is currently considerable interest in multi-metallic catalysis since it allows for the design of specifically homogeneous hetero-bimetallic catalysts that can facilitate the activation of different electrophiles through the stereoelectronic characteristics of two metals present in a single compound, thus promoting selective binding to a substrate. In this sense, the use of hetero-bimetallic catalysts constitutes an alternative method for the functionalization of propargyl alcohols. For example, using an Ir^III^-SnI^V^ catalyst in 1,2-dichloroethane (DCE) as a solvent enabled the activation of propargyl alcohols **63** (electrophiles), which reacted with a series of aromatic nucleophiles (Nu-H) **107** regioselectively, to furnish aryl-propargylated derivatives **108** with high turnover frequency (TOF) and with moderate to good yields (Figure 38, entry 1) [132]. Furthermore, the direct propargylation of arenes **107** with propargyl alcohols **63** was promoted by SnCl_2_ or Ce(OTf)_3_ in MeNO_2_ as a solvent. These transformations resulted in high selectivity toward the propargylated products **108** (Figure 38, entry 2 and entry 3) [133,134].

#### 2.3.3. With Propargyl Fluorides

The Nicholas reaction has been employed as an alternative to circumvent the challenges involved in the propargylation of arenes, but this method has drawbacks because it uses Co_2_(CO_6_), requires several steps, and gives low yields with electron-poor arenes. The ionization of propargyl fluorides **19** (X = F) in trifluoroacetic acid (TFA) in a mixture of DCM/HFIP as solvents produced products **108** in acceptable to excellent yields (Figure 39), thus providing a viable method to directly obtain a variety of substituted aryl-propargyl derivatives **108** in a Friedel–Crafts-type propargylation reaction [135].

#### 2.3.4. With Propargyl Phosphates

The copper-catalyzed direct propargylation of polyfluoroarenes **107** (n = 4 and 5) with secondary propargyl phosphates **109** that uses a strong base, such as, *t*BuOLi or THF, as a solvent has been described. Using this method, a series of propargylated polyfluoroarenes **108** were synthesized in moderate to good yields, with high chemo- and regioselectivity (Figure 40). Furthermore, this reaction could also be extended to triethylsilyl- and *tert*-butyl substituted alkynes [136].

#### 2.3.5. With Propargyl Cation Equivalents

Given the prevalence of the phenol motif in bioactive molecules, pharmaceuticals, and functional materials [137], a series of *ortho*-propargyl phenols **111** were synthesized via a boron-catalyzed sequential procedure through the addition of terminal alkynes **62** (R^2^ = Aryl) to substituted phenols **110**, bearing congested quaternary carbons (Figure 41). Control experiments combined with DFT calculations suggested that the reaction proceeds via a sequential phenol alkenylation/hydroalkynylation process [138].

(b)Heterocyclic derivatives

(i)Indoles

#### 2.3.6. With Propargyl Alcohols, Ethers, and Esters

*N*-Heterocyclic systems are important as building blocks of natural products, drugs, and functional organic materials, and the development of mild and selective methods for the direct introduction of propargyl groups into heterocyclic rings is highly desirable in order to access important and novel organic precursors.

Focusing on indoles, Table 4 provides a summary of available methods for the synthesis of propargyl–indole hybrids **113** via the reaction of indole derivatives **112** with diversely substituted propargyl derivatives **54**/**63**, employing various Lewis acids, zeolites, and superacids, in molecular solvents, as well in ionic liquids (entries 1-7) [134,139,140,141,142,143,144].

Enantioselective propargylation between indoles **112** and propargyl esters **54**, catalyzed by the transition metal CuOTf•1/2C_6_H_6_, was reported in the presence of a chiral ligand ((4*S*,5*R*)-diPh-Pybox) in 4-methylmorpholine and MeOH, leading to products **113** in moderate to high yields, (Table 4, entry 8) [145]. Likewise, an asymmetric procedure was described, consisting of Friedel–Crafts alkylation between substituted indoles **112** and propargyl carbonates **54**, in the presence of Ni(cod)_2_ and the chiral ligand (*R*)-BINAP and a base, in toluene, forming propargyl–indole derivatives **113** with high enantioselectivity and regioselectively and in moderate to good yields (entry 9) [146].

#### 2.3.7. With Allenyl Bromides

A direct method for a C-H propargylation reaction of indole derivatives **112** using bromoallenes **19c** (X = Br) was reported, which employed Mn(I)/Lewis acid as cocatalyst [147]. The presence of BPh_3_ not only promoted reactivity, but also enhanced selectivity. Using this method, secondary, tertiary, and even quaternary carbon centers in the propargylic position could be directly constructed, leading to diversely substituted propargyl–indoles **114** in moderate to high yields (Figure 42) [147].

(ii)Other heterocyclic substrates

#### 2.3.8. With Propargyl-TMS

The same approach as that described in Figure 37 was adopted for the direct metal-free *ortho*-propargylation of heteroaromatics **115** to produce *o*-propargylated heteroaromatic sulfides **116**. Thus, mixtures of thiophenyl or furanyl sulfoxide **115**, propargyl-TMS derivatives **17**, and Tf_2_O were reacted in ACN as a solvent to produce products **116** regioselectively and in good to excellent yields (Figure 43) [131].

Following the approach described in Figure 36, a method for the synthesis of *ortho*-propargyl iodothiophenes **119**/**120** was described [130]. In this case, a mixture of propargyl-TMS derivative **12**, thiophenyliodine diacetates **117**/**118**, and BF_3_•OEt_2_ in ACN/DCM as a solvent was allowed to react at low temperature to produce products **119**/**120** regioselectively, and in good yields (Figure 44) [130].

#### 2.3.9. With Allenyl Bromide

Following the procedure described in Figure 42, propargylated pyrrole and thiphene derivatives **125**–**128** were obtained in acceptable to good yields from bromoallenes **19c** (X = Br), and the corresponding heteroaromatic precursors **121**–**124** are shown in Figure 45 [147].

#### 2.3.10. With Propargyl Alcohols

Figure 46 gives an overview of the reported methods for the synthesis of propargylated heterocycles **134**–**139** using propargyl alcohols **63**. A wide variety of catalytic systems have been employed, including hetero-bimetallic catalysts of Ir^III^-SnI^V^ (entry 1) [132], Pd-Sn bimetallic catalysts (entry 2) [148], Ce(OTf)_3_ (entry 3) [134], and boron Lewis acids (entry 4) [149]. Doubly propargylated *N-*methylcarbazoles **136** were synthesized in [BMIM][PF_6_]/TfOH (entry 5) [150], and [BMIM][BF_4_]/Sc(OTf)_3_ proved effective for the propargylation of various classes of heterocycles under mild reaction conditions (entry 6) [151].

### 2.4. Acyl Halides

#### With Propargyl-Organolithium Reagent

Homopropargyl and *bis*-homopropargyl alcohols are convenient intermediates in organic synthesis [152]. Previous studies have established that the controlled lithiation of allenes forms operational equivalents of propargyl dianions (C_3_H_2_Li_2_, 1,3-dilithiopropyne) **143** [153,154]. In this vein, controlled dilithiation of propargyl bromide with two equivalents of *n*-butyllithium, in the presence of TMEDA, was reported to be a productive method for the synthesis of *bis*-homopropargylic alcohols **142** (Figure 47). In this approach, dianion **141** underwent in situ reactions with acid chlorides **140** to produce alcohols **142** in moderate yields with high regioselectivity [155].

### 2.5. Amine/Amide Derivatives

#### 2.5.1. With Propargyl Alcohols

Figure 48 gives an overview of the reported methods for the synthesis of *N*-propargylamines **97/144** from secondary propargyl alcohols **63**, utilizing SnCl_2_ in CH_3_NO_2_ (entry 1) [133] and Sc(OTf)_3_ in [BMIM][BF_4_] (entry 2) [151] as catalysts.

Figure 49 highlights an efficient tandem propargylation–cyclization–oxidation procedure for the synthesis of diversely substituted pyrimidines **147** via propargylamine intermediates **146**, by reacting propargylic alcohols **63** with amidine **145** using copper(II) triflate as a catalyst [156].

#### 2.5.2. With Propargyl Bromide

Among the nitrogen-containing fused heterocycles, quinoline, azepine, and triazole moieties are considered privileged scaffolds, are present in numerous natural products, and are among the most widely exploited heterocyclic rings for the development of bioactive molecules [157,158,159]. The propargylation of secondary amines **149**, prepared via the reductive amination of 2-chloro-3-formylquinolines **148**, produced tertiary propargylamines **150** as key intermediates for the synthesis of fused-heterocyclic products **151**, incorporating three active pharmacophores (quinoline, azepine and triazole) in a single molecular framework [160]; this illustrates the potential of the *N*-propargyl moiety in heterocyclic synthesis (Figure 50).

Chiral *N*-*tert*-butanesulfinyl imines are important for the stereoselective synthesis of nitrogen-containing heterocyclic systems [161]. With the goal of synthesizing 3-substituted 1,2,3,4-tetrahydroisoquinolines **153** in an enantioenriched form, the *N*-propargylation of enantioenriched homopropargylic amines **74** was performed under basic conditions to give the corresponding 4-azaocta-1,7-diyne intermediates **152** in fair to good yields (Figure 51). An oxidation step, followed by [2+2+2] cyclotrimerization promoted by a Wilkinson catalyst, produced the target structure **153** which contained substituents at the 3-, 6- and 7-positions in high yields [106]. This illustrative example highlights the efficacy of *bis*-homopropargylamine in heterocyclic synthesis.

The *N*-propargylation of vinyl sulfoximines **154** with propargyl bromide **19a** produced *N*-propargyl-sulfoximines **155** as highly functionalized biologically promising small molecules (Figure 52) [162].

The *N*-propargylation of substituted isatins **4** (R = H) was accomplished via a microwave-assisted reaction using anhydrous K_2_CO_3_ as base in DMF solvent, according to Figure 53, to produce a set of diversely substituted *N*-propargyl isatins **156** in good to excellent yields [163].

Similarly, a library of *N*-propargyl 4*H*-pyrano[2,3-*d*]pyrimidine derivatives **158** was prepared through the *N*-propargylation of pyrano derivatives **157,** under ultrasound-assisted reaction conditions via phase transfer catalysis, according to Figure 54 [164].

A procedure for the synthesis of a series of *N*-propargylated compounds **160a**–**f** was conducted, according to Figure 55 [165], using azazerumbone (**159a**), azazerumbone oxides (**159b,c**), acridin-9(10*H*)-one (**159d**), 7-methoxy-6-[3-(morpholin-4-yl)propoxy]quinazolin-4(3*H*)-one (**159e**), and murrayafoline A (**159f**) as substrates.

A series of nucleobase derivatives **165**–**168** were synthesized via the propargylation of DNA nucleobases **161**–**164** according to Figure 56, with the goal of extending their functionality to obtain biofunctional materials. The in vitro biocompatibility of the native **161**–**164** and nucleobase derivatives **165**–**168** was assessed using primary human dermal fibroblasts (HF), showing that they were non-toxic, and hence, suitable for biomedical applications [166].

#### 2.5.3. With Propargylic Cation Intermediates

The nucleophilic addition of the primary amino-ester **169** to cobalt-stabilized propargylic carbocation **170**—initially in the presence of BF_3_•OEt_2_, followed by CAN, as catalytic systems—generated the corresponding dipropargylamino-ester **171** according to Figure 57 [167].

### 2.6. Vinylstananes

#### With Propargyl Bromide

A methodology involving the coupling of vinyl-stannanes (β-trifluoromethyl (*Z*)-α- and (*Z*)-β-stannylacrylates) **172** to propargylic bromides **19a** catalyzed by copper(I) provided access to the corresponding propargylated products **173** without allenic transposition (Figure 58). This Pd-free cross-coupling process tolerated various R-groups, and occurred with retention of the configuration at the double bond; furthermore, homocoupling and allenic products were not detected [168].

### 2.7. (a) Alcohols, (b) Enol-Like Precursors, (c) Phenols, (d) Thiols, and (e) Carboxylic Acids

(a)Alcohols

#### 2.7.1. With Propargyl Bromides

The propargylation of hydroxyl-amides **174**, synthesized via a Passerini reaction mediated by boric acid, generated *O*-propargyloxyamides **175** as key intermediates (Figure 59) [169], whose cyclization in the presence of potassium *tert*-butoxide via a 5-*endo*-*dig* process produced a series of 2,5-dihydrofurans **176** of synthetic interest [170,171,172,173].

Expanding on the strategy for the synthesis of quinoline/azepine pharmacophores fused to a triazole moiety (see Figure 50), hetero-polycyclic products **179** were obtained from (2-chloroquinolin-3-yl)methanol derivatives **177** via the *O*-propargylation of **177** to give the key propargyl intermediates **178**, followed by a click reaction and Pd-catalyzed C-H functionalization (Figure 60) [160].

The *O*-Propargylation of oxime **180** with propargyl bromide **19a**, according to Figure 61, provided facile access to the perylenediimide compound **181**, whose main characteristic was its capability to detect Cu^2+^ and Pd^+2^ ions in water [174].

Figure 62 highlights two synthetic strategies for access to propargylated ethers **183** and **186**. The first process involves the cyclization of L-glutamic acid to obtain the lactone **182**, which was reacted with propargyl bromide **19a** in alkaline medium in a mixture of polar aprotic solvents to obtain the propargylated lactone **183** in moderate yields [175]. Compound **183** was then used as a starting point for multistep synthesis, leading to polycyclic compound **184**. The goal of the second etherification process was to generate propargylated disaccharides. In this case, glycoside **185** was reacted with propargyl bromide **19a** to produce the tetra-propargylated arabino-3,6-galactane **186** in good yields [176].

Figure 63 highlights a method for the synthesis of terminal *gem*-difluoropropargyl ethers **190** from *gem*-difluoropropargyl bromide dicobalt complex **188** in the presence of silver triflate and TEA in toluene. Complex **188** reacted selectively with aliphatic alcohols **187**, even if the substrates **187** contained other nucleophilic functional groups, producing propargyl ether complexes **189**. Decomplexation of the resulting dicobalt complexes **189** using cerium ammonium nitrate (CAN) or *N*,*N*,*N*′-trimethylethylenediamine, followed by desilylation by TBAF, produced compound **190** [177].

Implementing the strategy outlined in Figure 55, a series of *O*-propargylated compounds **191a-d** bearing one or two propargyl groups in their structures were synthesized using 3-methyl-9*H*-carbazol-1-ol (**187a**), 4-hydroxycoumarin (**187b**), and α-mangostin (**187c**) as substrates (Figure 64). These compounds were evaluated for their in vitro cytotoxicity against three human cancer cell lines, the HepG2, LU-1, and Hela cell lines. Compound **191c** proved most active, showing IC_50_ values of 1.02, 2.19, and 2.55 μg/mL, respectively [165].

#### 2.7.2. With Propargyl Esters

Compounds **194/195** and **196** were synthesized via *O*-propargylation of the monosaccharide **194** and hydroxylic precursors **193** with propargyl esters **54**, employing dual catalysis between [Cu(ACN)_4_]BF_4_ and boronic acid (**B**), and using a chiral ligand ((***S***,***S***)-**L**) in the presence of a weak base (TEA) in THF (Figure 65). A notable feature of this approach is the formation of several stereocenters in a chemo- and stereoselective manner [178,179].

#### 2.7.3. With Propargyl Alcohol/Ethers

An efficient method for the synthesis of end-functionalized oligosaccharides from unprotected monosaccharides using a one-pot/two-step approach was developed (Figure 66) [180]. In the first step, mannose **197** was functionalized with propargyl alcohol **63** (R = R^1^ = H) at the anomeric position through Fisher glycosylation using Amberlyst-15, producing a propargyl monosaccharide **198**. In a second step, the reaction mixture was heated under vacuum at 100 °C in order to increase the degree of polymerization of **198**, leading to a fully functionalized propargylated glycoside **199**, with a degree of polymerization (n) up to 8 [180].

Propargyl ethers **200** were synthesized by reacting propargylic alcohols **63** and different primary and secondary alcohols **187** in the presence of catalytic amounts of aqueous HBF_4_ as a catalyst (Figure 67) [181].

Implementing the procedure described in Figure 57, the corresponding propargylated amino-ethers **203** were synthesized via a reaction of dicobalt hexacarbonyl-complexed (Co_2_(CO)_6_)-propargyl methyl ether **202** with aminoalcohols **201** in the presence of BF_3_•OEt_2_ and CAN as catalytic systems (Figure 68) [167].

(b)Enolic substrates

#### 2.7.4. With Propargyl Bromides

The reaction of difluoropropargyl–bromide–dicobalt complexes **188** with enolizable ketones and aldehydes **204**, in the presence of AgNTf_2_ and with *i*Pr_2_NEt or DTBMP as a base, led to the synthesis of difluoropropargyl vinyl ether–dicobalt complexes **205** bearing diverse substituents (Figure 69). These compounds were then utilized as convenient precursors for the synthesis of difluorodienone and difluoroallene derivatives [182].

(c)Phenolic substrates

#### 2.7.5. With Propargyl Bromides

The propargylation of phenolic hydroxyl groups is important because of its potential as starting material for the preparation of high-molecular-weight synthetic and natural polymers. The reaction of propargyl bromide **19a** with the phenolic OHs of the lignin derivative **206**, in the presence of an aqueous base, yielded a propargylated-lignin product **210** (entry 1) [183]. In other studies, the propargylation of phenols **207**, **208**, and **209**, in the presence of K_2_CO_3_ as catalysts in acetone or DMF and under MW irradiation, produced the corresponding propargylated ethers **211** (entry 2) [184], **212** (entry 3) [185], and **213** (entry 4) [186] (Figure 70). These compounds were further functionalized via “click” chemistry.

#### 2.7.6. With Propargyl Alcohols/Ethers

Following the procedure described in Figure 57, propargylated tyrosine derivatives **215**, were prepared starting from with dicobalt complexes **202** as propargylating agents, according to Figure 71, and employing BF_3_•OEt_2_ and CAN as catalytic systems [167].

(d)Thiolic substrates

#### 2.7.7. With Propargyl Bromide

Thiobenzimidazole- **216** and cysteine-containing peptides **217** were *S*-propargylated using a mild base, according to Figure 72, to produce propargylated thiobenzimidazole **218** (entry 1) [187] and propargylated peptides **219** (entry 2) [188].

#### 2.7.8. Propargylic Cation Intermediates

*S*-propargylated cysteine ethyl ester derivatives **221** were prepared according to the conditions established in Figure 57, starting with propargyl–dicobalt complexes **170** in the presence of BF_3_•OEt_2_ and CAN as catalytic systems (Figure 73) [167].

(e)Carboxylic acids

#### 2.7.9. With Propargyl Bromide and Propargylamine

The propargylamides **224** were synthesized through a reaction between indoloacids **224** with propargylamine **222** (R = NH_2_) via an acyl chloride intermediate (generated in situ by reacting **223** with oxalyl chloride) (Figure 74, entry 1) [189]. Using the same approach, propargylation of natural maslinic acid **225** with propargyl bromide **19a** (R = Br) produced the desired propargyl derivative **226** (entry 2) [190].

The preparation of *C*-propargylic esters **228** was carried out via a reaction between *N*-protected amino acids **227** and propargyl bromide **19a** (R = Br) in DMF in the presence of anhydrous potassium carbonate (Figure 74, entry 3) [191].

#### 2.7.10. With O-Propargylated Hydroxylamine

A novel bio-orthogonal prodrug **231** of the HDACi panobinostat was developed that was harmless to cells and could be converted back into the cytotoxic panobinostat via Au catalysis. The key propargylated product **231** was obtained from *O*-propargylated hydroxylamine **230** with β-substituted-acrylic acid **229** using *N*-(3-dimethylaminopropyl)-*N*′-ethylcarbodiimide hydrochloride (EDC) in H_2_O, according to Figure 75 [192].

#### 2.7.11. With Propargylic Cation Intermediates

Following a similar procedure to that described in Figure 57, the propargylated *N*-Bz-D-phenylalanine **232** was synthesized through its carboxyl–CO_2_H functionality, by reacting the propargyl–dicobalt complex **170** with a phenylalanine derivative **227** (R^1^ = Bn) in the presence of BF_3_•OEt_2_ and CAN (Figure 76) [167].

### 2.8. (a) Alkenes, (b) Allenes, and (c) Enynes

(a)Alkenes

#### 2.8.1. With Propargyl-/Allenylboron

Catalytic enantioselective allylic substitution is a widely used strategy in organic synthesis, because it transforms an alkenyl substrate into a new unsaturated compound bearing an allylic stereogenic center [193].

Transformations of acyclic, or aryl-, heteroaryl-, and alkyl-substituted penta-2,4-dienyl phosphates **233**, as well as cyclic dienyl phosphates **234**, were carried out in the presence of commercially available allenyl-*B*-(pinacolato) **2c**, mediated by a sulfonate-containing NHC-Cu complex (NHC = imidazolyl carbene). Products **235/236** were obtained that contained, in addition to a 1,3-dienyl group, a readily functionalizable propargyl moiety (Figure 77). The positive attributes of this reaction were high yields, high *E:Z* ratios, and impressive enantiomeric ratios (*er*). Kinetic isotope effect measurements and DFT computations provided mechanistic insights into this catalytic process [194].

Focusing on allylic substitution, in another study, 1,5-enynes **238** were synthesized via a silver-catalyzed allylic substitution by reacting a propargylic organoboron compound **2a** with allylic phosphates **237**, using a chiral *N*-heterocyclic carbene (NHC) ligand and a silver catalyst complexed to a copper chloride salt (Figure 78) [195]. In all cases, the incorporation of the propargylic group was favored over allenyl addition.

#### 2.8.2. With Propargyl Alcohols

The 1,5-enynes **240** were synthesized via the reaction of allyltrimethylsilane **239** with propargylic alcohols **63** in the presence of Bi(OTf)_3_ in [bmim][BF_4_] ionic liquid (IL) (Figure 79). The reaction exhibited a broad substrate scope, with the possibility for the recovery/reuse of the IL solvent with a minimal decrease in isolated yields, after six cycles [196].

In another approach, diarylalkenyl propargylic frameworks **242** were synthesized via an Fe-catalyzed reaction of propargylic alcohols **63** with various symmetric and asymmetric 1,1-diarylethylenes **241** (Figure 80). The reaction worked well for a wide range of ethylenes **241** bearing electron-donating or electron-withdrawing groups (as R^2^ or R^3^ substituents) [197].

An efficient catalytic method for the propargylation of quinones **243** that benefits from the cooperative effect of Sc(OTf)_3_ and Hantzsch ester (HE) has been reported, yielding the corresponding propargylated quinone derivatives **244** (Figure 81). Using this approach, a broad range of propargylic alcohols **63** were converted into the appropriate propargyl derivatives **244** in acceptable to excellent yields [198].

#### 2.8.3. With Propargyl Bromides

The development of enantioselective alkyl–alkyl cross-couplings with the formation of a stereogenic center is significant and highly desirable. In this context, the regio- and enantioselective Ni-catalyzed hydropropargylation of acrylamides **245** yielded propargylamides **246** bearing a tertiary stereogenic carbon center (Figure 82). This protocol was carried out using propargyl bromides with alkyl, aryl, and silyl substituents **19a** in the presence of a NiBr_2_ glyme, an (*R*,*R*)-**L12** chiral ligand, trimethoxylsilane, potassium phosphate monohydrate, and *tert*-butanol in diethyl ether, producing Csp^3^–Csp^3^ cross-coupling products **246** in good yields and with excellent enantioselectivities [199].

(b)Alkenes

#### 2.8.4. With Propargyl Ethers/Esters

Allenamides have received increasing attention in recent decades due to their diverse reactivity. In this context, highly diastereoselective oxy-propargylamination of allenamides **248** with *C*-alkynyl *N*-Boc-acetals as difunctionalization reagents **247** has been described, which employs XPhosAu-(MeCN)PF_6_ as a catalyst. This methodology provided highly functionalized propargyl-1,3-amino alcohol derivatives **249** in acceptable to good yields and with good to excellent diastereoselectivities (Figure 83) [200].

#### 2.8.5. With Propargyl Bromides

A series of (*E*/*Z*)-3-amidodienynes **251** were synthesized via a tandem α-propargylation–1,3-H isomerization reaction of chiral allenamides **250** and propargyl bromides **19a** with moderate *E*/*Z* ratios. Subsequently, the reactivities of these *E*/*Z*-isomers **251** were examined via thermal Diels–Alder cycloaddition reactions. The results showed that only the (*Z)*-3-amidodienynes (***Z***)-**251** reacted to provide *endo-II* products **253** (Figure 84) [201].

(c)Enynes

#### 2.8.6. With Propargyl Alcohols

The chemoselectivity in the 1,4-carbooxygenations of 3-en-1-ynamides **254** with propargylic alcohols **63** was examined using a gold catalyst via non-Claisen pathways. The reactions were performed with electron-rich propargylic alcohols **63**, using Ph_3_PAuCl/AgOTf as a catalytic system in toluene, producing 1,4-oxopropargylation products **255** in good yields and with high *E*-selectivity (Figure 85) [202].

A chiral ruthenium-based complex was prepared from (TFA)_2_Ru(CO)(PPh_3_)_2_ and (*R*)-BINAP in order to catalyze the enantioselective C−C coupling of diverse-type primary alcohols **187** with conjugated enyne **60**. This approach produced secondary homopropargyl alcohols **256** bearing *gem*-dimethyl groups in their structures (Figure 86) [203].

### 2.9. Carbanionic-Like Nucleophiles

#### 2.9.1. With Propargyl Alcohols

Propargylations of 1,3-diketones **257** were achieved with propargylic alcohols **63** mediated by Lewis and Brønsted acidic ILs in the presence of the metallic triflate Sc(OTf)_3_ or Bi(NO_3_)_3_ as catalysts, and produced products **258** (Figure 87, entry 1). The scope of this condensation reaction was investigated using a variety of propargylic alcohols and a host of β-ketoesters **259** and cyclic dicarbonyl compounds **260**, producing the corresponding adducts **261** and **262**, respectively. The [BMIM][PF_6_]/Bi(NO_3_)_3_•5H_2_O catalytic system proved superior for propargylation reactions, and the IL solvent could be recycled and reused [204].

Using Sc(OTf)_3_ as catalysts, alkynyl diesters **264** were synthesized via propargylations of 1,3-diesters **263** using 3-sulfanyl and 3-selanylpropargyl alcohols **63** (R^1^ = SPh, SePh) in MeNO_2_–H_2_O. Cyclic alkynyl diketones **265** and ketoesters **266** were similarly propargylated, (Figure 87, entry 2). Further, under the action of bases such as Bu_4_NF, CsCO_3_, K_2_CO_3_ and NaH, some of the obtained propargylated derivatives **264**, **267**–**268** underwent intramolecular cyclization to give diversely substituted tetrahydro-benzofurans [205].

Propargylic alcohols can be activated towards S_N_1-type reactions with nucleophiles using a variety of Lewis acids or Brønsted acids as catalysts [206]. In this process, the highly stereoselective organocatalytic alkylation of internal propargylic alcohols with aldehydes has been described, with water used as a solvent, using a mixture of In(OTf)_3_ and the MacMillan organocatalyst **L***; these worked in a cooperative manner to produce propargyl aldehydes **270** regioselectively (Figure 88). The reported method is versatile and tolerates diverse functional groups, allowing for the use of highly functionalized internal alkynes **63** and aldehydes **269** as precursors. According to the reaction conditions, the formation of **270** proceeds via an S_N_1-type reaction involving a stabilized propargylic cation species formed via the ionization of propargylic alcohols **63** [207].

Expanding on propargylation reactions mediated by Lewis and Brønsted acidic ILs (in Figure 87), a [BMIM][PF_6_]/Bi(NO_3_)_3_•5H_2_O catalytic system proved efficient for the propargylation of 4-hydroxycoumarins **187b**, producing the corresponding propargylated 4-hydroxycoumarins **271** (Figure 89) [204].

#### 2.9.2. With Propargyl Halides/Phosphoesters

With the goal of synthesizing the bicyclic fragment (i.e., AE rings) of the *Daphniphyllum* alkaloid yuzurine, the key intermediate **272** was synthesized via the diastereoselective propargylation of the α-position of lactone **271** with propargyl bromide **19a** (X = Et) (Figure 90, entry 1) [208]. In other approach, the propargylation of Ugi adducts **273** with propargyl bromide **19a** (X = H), under the addition of excess sodium hydride in DMSO, led to the direct formation of pyrrolidinone enamides **275**. Products **275** were produced via the intermediate formation of the propargyl derivatives **274**, and cyclized in situ through the action of NaH (Figure 90, entry 2). The latter compounds **275** were identified as useful precursors of iminium intermediates, and were applied to the formation of benzoindolizidine alkaloids via Ugi/propargylation/Pictet−Spengler cyclization [209].

1,3-diester **276** was propargylated with propargyl bromide **19a** (X = H) using metallic zinc in DMF, producing the corresponding propargyl 1,3-diester **277** (Figure 90, entry 3) [210]. In the context of multistep asymmetric total synthesis, the propargyl intermediate **279** was synthesized in a highly stereoselective fashion via LDA-mediated propargylation of the 1,3-dioxolanone **278** with propargyl bromide **19a** (X = H), producing intermediate **279** (Figure 90, entry 4) [211].

With the aim of evaluating the influence of ultrasound in association with a new phase-transfer catalyst (PTC) for synthetic purposes, 2,2-di(prop-2-ynyl)-1*H*-indene-1,3(2*H*)-dione **281** was synthesized via the propargylation of indene-1,3-dione **280** with propargyl bromide **19a** (X = H) using aqueous potassium hydroxide under phase-transfer catalysis, employing *N*-benzyl-*N*-ethyl-*N*-isopropylpropan-2-ammonium bromide and ultrasonic irradiation in chlorobenzene (Figure 90, entry 5). Based on a kinetic study, it was established that the overall reaction rate can be greatly enhanced with ultrasound irradiation [212].

Figure 91 illustrates the reported synthesis of γ-ketoacetylene **284** via a condensation reaction between propargyl chloride **282** and β-keto ester **283** in the presence of sodium hydride [213]. This compound is a key intermediate in the biomimetic synthesis of plumarellide, a polycyclic diterpene [214].

1,4-Diynes are valuable and versatile synthons for natural products, organometallic complexes, and the synthesis of novel molecules [215]. Figure 92 illustrates a reported method for the catalytic synthesis of difluorinated compounds **286**, difluoromethylene (CF_2_)-skipped 1,4-diynes, via palladium-catalyzed cross-coupling between terminal alkynes **62** and *gem*-difluoropropargyl bromide **285** in toluene. The method exhibited high functional group tolerance and a broad substrate scope [216].

Compounds bearing a quaternary carbon stereocenter are important building blocks in medicinal chemistry, and are found in biologically active compounds such as pharmaceuticals and agrochemicals. Figure 93 illustrates an efficient enantioselective method for the asymmetric α-alkylation of α-branched aldehydes **204** with propargyl bromide **19a** to generate products **287** bearing a chiral quaternary carbon stereocenter. The reaction proceeds through enamine-based organocatalysis using a chiral primary amino acid as a catalyst [217].

Propargylated products **289** were synthesized via the Suzuki-type coupling of propargylic electrophiles **19d/109** with diborylmethane **288**, using CuI/PPh_3_ as the catalytic system and *t*BuOLi as a base, under mild conditions with good functional group tolerance (Figure 94) [218].

#### 2.9.3. With Propargyl Ethers or Esters

The diastereo- and enantioselective synthesis of 2,2-disubstituted benzofuran-3(2*H*)-ones **291** was achieved via a “copper-pybox”-catalyzed reaction between 2-substituted benzofuran-3(2*H*)-ones **290** and propargyl acetates **200** (R = Ac), as outlined in Figure 95, entry 1. The positive attributes of the method were good functional group tolerance and broad substrate scope. The utility of the method was demonstrated by further transformation of the terminal alkyne of **291** into a methyl ketone without loss of enantiomeric purity [219]. Using a similar approach, propargyl tricarboxylate derivatives **293** were synthesized via the copper-catalyzed enantioselective propargylation of triethylmethanetricarboxylate **292** with propargylic alcohol derivatives **200**. The active catalyst “copper-pybox” was generated by combining the copper complex Cu(CH_3_CN)_4_BF_4_ with (*S*)-*sec*-butyl-Pybox (Ligand **L1***) at low temperatures in methanol, with DIPEA as base, as outlined in Figure 95, entry 2. The scope of the methodology was demonstrated using phenyl-substituted propargylic substrates **200** bearing electron-donating as well as electron-withdrawing groups at the *para*-position of the phenyl ring [220].

The efficacy of the copper–ligand complexes in stereoselective synthesis with propargyl esters are showcased here with the following examples, sketched in Figure 96:

(i)The synthesis of a series of optically active 3,3-disubstituted oxindole skeletons **295** bearing vicinal tertiary and all-carbon quaternary stereocenters via the propargylation of 3-substituted oxindoles **294** with propargylic acetates **200**, using Cu(ACN)_4_PF_6_ combined with a chiral tridentate ferrocenyl, *P*,*N*,*N*-ligand **L1***, in methanol, entry 1 [221].(ii)The synthesis of a series of propargyl nitro derivatives **297** bearing two contiguous stereogenic centers by reacting propargylic carbonates **200** with α-substituted nitroacetates **296** using Cu−pybox as catalyst. The most striking features of these reactions are the observed high diastereo- and enantioselectivities. Products **297** were further employed as precursors of non-proteinogenic quaternary α-amino acids after the reduction of their nitro groups, entry 2 [222].(iii)The synthesis of highly functionalized chiral propargylated *P*-ylides **299** via the copper-catalyzed asymmetric propargylation of phosphonium salts **298** with racemic propargylic esters **200**, in the presence of the chiral ligand **L***, and further Wittig reactions of **299** with aliphatic aldehydes; this led to the synthesis of diversely substituted chiral propargylated alkene building blocks **300** (Figure 96, entry 3), with a wide substrate scope and satisfactory functional group compatibility [223].(iv)The synthesis of terminal alkyne-containing products **303** and **304** bearing two vicinal stereocenters via an asymmetric propargylic substitution (APS) reaction of thiazolones **301** (A = S) and oxazolones **302** (A = O) with propargyl esters **200** (X = H) mediated by Cu/Zn and Cu/Ti dual metal catalytic systems (Figure 96, entry 4). The resulting functional group-rich products exhibited good to excellent diastereo- and enantioselectivities [224].(v)The enantioselective synthesis of propargylic diesters **305** via a nickel/Lewis acid-catalyzed asymmetric propargyl substitution, by reacting achiral starting-type materials **263** and **54** under mild conditions. The introduction of a Lewis acid cocatalyst such as Yb(OTf)_3_ was crucial in transforming the mixture of **263** and **54** into products **305** (Figure 97). Further, this asymmetric propargylic substitution reaction was investigated for the development of a range of structurally diverse natural products and seven biologically active compounds, namely, (−)-thiohexital, (+)-thiopental, (+)-pentobarbital, (−)-AMG 837, (+)-phenoxanol, (+)-citralis, and (−)-citralis, demonstrating the efficacy of this asymmetric strategy [225].(vi)Enantioselective copper-catalyzed vinylogous propargylic substitution with coumarin derivatives. In this approach, aromatic and aliphatic propargylic esters **200** reacted with substituted coumarins **306** under mild conditions to yield propargylated coumarin derivatives **307** with impressive enantioselectivities (Figure 98). Further, biological studies on the compounds **307** led to the discovery of a novel class of autophagy inhibitors [226].

A catalytic system based on *bis*(triphenylphosphine)palladium (II) dichloride, Ag_2_CO_3_, and phosphine-based ligand **L** was developed for the one-pot selective synthesis of diversely substituted dihydrofuro[3,2-*c*]coumarins **308**. The synthetic strategy involved a propargylation reaction between propargylic carbonates **54** and 4-hydroxycoumarins **187b**, mediated by the aforementioned catalytic system (Figure 99). Mechanistic studies have suggested that 4-hydroxycoumarins **187b** react with an *η*^1^–(propargyl)palladium complex, formed in situ, to generate the key terminal alkyne intermediate **271**, which undergoes selective intramolecular 5-*exo*-*dig* cyclization to give the isolated products **308** in one pot [227].

A series of substituted pyrrole derivatives **310** were synthesized via a zinc(II) chloride-catalyzed regioselective propargylation/amination/cycloisomerization process by reacting enoxysilanes **309** with propargylic acetates **200** and primary amines **94**. This method was applicable to a variety of aromatic and aliphatic propargylic acetates **200** without the necessity of isolating intermediates such as **258** (Figure 100) [228].

A series of diversely substituted propargyl ethers **311** were obtained via a Re(I)-catalyzed hydropropargylation reaction between silyl enol ethers **309** and propargyl ether **191** (Figure 101). Mechanistic studies suggested that the reaction proceeded via the intermediacy of vinylidene–alkenyl metal intermediates undergoing a 1,5-hydride transfer to generate the isolated products **311** [229].

Fully substituted pyrroles are important bioactive motifs, and are widely presented in many biologically active compounds and natural products [230]. In this context, a copper-catalyzed and microwave-assisted tandem propargylation/alkyne azacyclization/isomerization sequence between propargyl acetates **200** and β-enamino compounds **312** was established (Figure 102). Through this process, a series of pentasubstituted pyrroles **314** were synthesized. This transformation was characterized by a broad substrate scope that tolerated diverse substituents in its starting materials **200** and **312**, and could be scaled up for further biomedical research. A mechanistic sequence in which an enyne-like structure **313** acts as a key intermediate in the catalytic cycle was proposed [231].

A highly diastereo- and enantioselective method for the synthesis of compounds **316/317** bearing vicinal tertiary stereocenters was devised by reacting propargylic acetates **200** with morpholine-derived cyclic enamine **315**, in the presence of a copper catalyst, a chiral tridentate *P*,*N*,*N*-ligand ((***R***)**-L***), and *i*Pr_2_NEt in MeOH. This approach was compatible with a wide range of substrates **200**, producing chiral propargylated cyclohexanones **316/317** in good yields and with excellent diastereoselectivity (Figure 103) [232].

#### 2.9.4. With 1,3-Diarylpropynes

Direct C–C coupling from Csp^3^–H bonds with molecular oxygen as the terminal oxidant continues to be a challenging task. In this context, diversely substituted propargyl adducts **318** were synthesized via a coupling reaction between 1,3-dicarbonyl compounds **257/259** and 1,3-diarylpropynes **57** in the presence of molecular oxygen, DDQ, and sodium nitrite (Figure 104). The addition of HCO_2_H dramatically increased the speed of the process [233].

#### 2.9.5. With Propargyl Aldehydes

The metal-free, amino acid-catalyzed, three-component reductive coupling of propargyl aldehydes **319** and cyclic/acyclic methylene-active compounds **320/321**, in the presence of Hantzsch ester and (*S*)-proline as catalysts, produced diversely substituted and gram-scalable propargylated cyclic/acyclic systems **322/323** (Figure 105). To demonstrate the synthetic value of this protocol, in selected cases, adducts **322/323** were further transformed into dihydropyran derivatives through an annulative etherification reaction using AgOTf as a catalyst [234].

The propargylated alcohol **325** was synthesized via catalytic asymmetric propargylation of the highly enolizable β-keto-lactone **324** with propargyl aldehyde **319** (Figure 106). The reaction was mediated by an Evans aldol type reaction [235], promoted by rigorously acid-free Sn(OTf)_2_. Notably, the synthesis of this compound was a key step in the total synthesis of leiodermatolide, a natural product derived from a deep-sea sponge with potent cytotoxic activity (Figure 106) [236].

### 2.10. Carbocationic Electrophiles

#### With Propargyl Organometallic-Based Reagents

A series of diversely substituted *o*-propargylated phenols **327** were obtained through the transition metal-free alkynylation of substituted 2-(tosylmethyl)phenols **326** with bromo(alkynyl)zinc reagents **89**, generated from the corresponding terminal alkyne with BuLi and ZnBr_2_, under N_2_ at room temperature. This efficient strategy exhibited good functional group compatibility (Figure 107). The products were further used as intermediates for the synthesis of 2,3-disubstituted benzofurans [237].

A method for the synthesis of spiroketals **329** bearing a five-membered and a seven- or eight-membered ring was described. In this approach, initially, the alkyne **328** was treated with Co_2_(CO)_8_ in DCM at room temperature to form the corresponding alkyne–Co_2_(CO)_6_ complex intermediates, which were subsequently exposed to BF_3_•OEt_2_ at low temperature to produce the desired dioxaspiro[4.7]-compounds **329** (Figure 108). This method was applicable to cyclopropanes possessing *gem*-disubstituents, as well as mono-aryl substituents [238].

The synthesis of a series of propargylic and homopropargylic alcohols **331/332** was accomplished via the reaction of epoxides **330** with 3,3,4,4-tetraethoxybut-1-yne acetylide **89** (M = Mg). The use of a MgBr counterion in the acetylide proved superior for the selective formation of propargylic alcohol **331**, while the use of a lithium acetylide and BF_3,_ followed by hydrolysis, gave homopropargylic alcohols **332** (Figure 109) [239].

### 2.11. Free-Radical-like Precursors

#### 2.11.1. With Propargyl Halides

Among the metal catalysts that promote alcohol C-H functionalization via C-X bond reductive cleavage pathways, rhodium-based catalysts were shown to be promising candidates [240]. In this sense, the carbinol *C*-propargylation of alcohols **187** with propargyl chlorides **19d** in basic media, under rhodium-catalyzed transfer hydrogenation, enabled the direct conversion of primary alcohols **187** into propargylated alcohols **13**. Interestingly, this methodology tolerated benzylic and heteroaromatic benzylic alcohols, as well as aliphatic and allylic alcohols **187**, producing the expected homopropargyl alcohols **13** in good yields (Figure 110) [241].

A radical hydrodifluoropropargylation method in which alkenes **241** are reacted with silyl-protected bromodifluoropropyne **285** in DMF, at room temperature and under irradiation with blue LEDs, has been described [242]. The method employed diphenyldisulfide and benzothiazoline **333** as reductants, yielding silyl-protected difluoropropargylated products **334** in acceptable to good yields, with wide functional group tolerance (Figure 111) [242].

#### 2.11.2. With 1,3-Enynes

The 1,3-enyne moiety has been recognized as an alternative pronucleophile for the carbonyl propargylation process [243]. Radical carbonyl propargylation via dual chromium/photoredox catalysis was recently reported [244]. Using this approach, a library of homopropargylic alcohols **336** bearing all-carbon quaternary centers was synthesized (Figure 112) via the catalytic radical tricomponent coupling of 1,3-enynes **60** (R^2^ = Me, CH_2_OH), aldehydes **1**, and suitable radical precursors (Hantzsch ester) **335** in the presence of an iridium-based photocatalyst (PC). This redox-neutral multi-component reaction occurred under mild conditions and showed high functional group tolerance, producing products **336** with acceptable diastereomeric ratios [244].

### 2.12. Boronic Acids (ArB(OH)_2_)

#### With Propargyl Bromides

The efficient microwave-assisted (MW), two-step synthesis of *N*-aryl propargylamines **144** from aromatic boronic acids **337**, aqueous ammonia, and propargyl bromide **19a** was reported. The first step involved copper-catalyzed coupling of aromatic boronic acids **337** with aqueous ammonia, which reacted with propargyl bromide **19a** in the second step to give a propargylamine derivative **144** (Figure 113, entry 1) [245]. In another approach, *gem*-difluoropropargyl derivatives **190** were prepared via the difluoropropargylation of boronic acids **337** with *gem*-difluoropropargyl bromide **285**, by employing [Pd_2_(dba)_3_]/P(*o*-Tol)_3_ (**L1**) as a catalyst in the presence of K_2_CO_3_ in dioxane (Figure 113, entry 2) [246].

### 2.13. Nitrones

#### With Propargyl Bromide

The propargylation of chiral nonracemic mono- and poly-hydroxylated cyclic nitrone derivatives **338**–**340** with Grignard reagents (generated in situ) was established as an efficient method for preparing building blocks containing an alkyne moiety **341**–**343**. These compounds were then employed in copper-catalyzed azide alkyne cycloaddition click chemistry [247]. The synthesis of **341**–**343** was accompanied, in most cases, by the formation of diastereomeric mixtures, and also required the use of (trimethylsilyl)propargyl bromide **19a** as a precursor for the formation of the Grignard reagent, in order to avoid the formation of undesired allene derivatives (Figure 114).

## 3. Conclusions and Outlook

This review has underscored the importance of the propargyl moiety as a highly versatile and powerful building block in organic synthesis. Propargylic and homopropargylic reagents have been synthesized from a variety of precursors and applied to a highly diverse array of substrates to synthesize propargylated derivatives. Judicious selections of catalysts, co-catalysts, and chiral ligands have resulted in the development the stereo- and enantio-selective synthesis of numerous functional small molecules, with applications in natural products and medicinal chemistry. The progress in this area during the last decade has been nothing short of astonishing. Clearly, this is a highly dynamic and continuously evolving research area, and we are confident that it will continue to advance in the coming decade.

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
