# Peer review of "Recent Advances in the Synthesis of Propargyl Derivatives, and Their Application as Synthetic Intermediates and Building Blocksâ€"

_molecules, 2023, doi:10.3390/molecules28083379_

Round 1
Reviewer 1 Report
In this manuscript, the author reports the recent advances in the synthesis of propargyl derivatives and their application as synthetic intermediates and building blocks in organic synthesis. The authors covered all the reports for propargylation agents and their application in the synthesis from 2010. The authors organized in logical order and explanation well. This review article would be an excellent source for young researchers and a quick overview of the propargylation reactions. With this, I recommend this manuscript for publication in Molecules. Thiers could respond to the below comments.
The 3Å and 4Å MS in Scheme 3., Table 3 needs to be re-written.
The author could provide the Fn for Scheme 40.
Scheme 63. Equiv. need to cross-check.
Authors could provide the reference for entries 2,4 for Table 2.
It could be more informative if authors could provide the publication years in the text.
Reviewer 2 Report
Propargyl group is a multi-reactive points functional group in organic compounds, which has been well-used as the building blocks in organic synthesis. This review summarized in the synthesis of propargyl derivatives, and their application as synthetic intermediates, it is interesting to the organic chemists, and the readers of Molecules. I recommend its publication in Molecules after revision subjected to the following points.
Several important review papers on the applications of propargyl compounds are omitted, which should be cited in the manuscript.
(a) Acc Chem Res. 2011, 44, 111
(b) Tetrahedron Lett. 2015, 56, 283.
(c) Mini-Reviews in Organic Chemistry, 2018, 15, 198
Reviewer 3 Report
This is a comprehensive review article of synthesis of propargyl derivatives and their synthetic application. Related papers which were published since 2010 have been collected. This informative review is useful for researchers.
